# Association of Drinking Herbal Tea with Activities of Daily Living among Elderly: A Latent Class Analysis

**DOI:** 10.3390/nu15122796

**Published:** 2023-06-19

**Authors:** Liyuan Tao, Jiaojiao Liao, Rui Zheng, Xiaoyu Zhang, Hongcai Shang

**Affiliations:** 1Key Laboratory of Chinese Internal Medicine of Ministry of Education and Beijing, Dongzhimen Hospital, Beijing University of Chinese Medicine, Beijing 100700, China; tendytly@163.com (L.T.); zhengr35@mcmaster.ca (R.Z.); 2Research Center of Clinical Epidemiology, Peking University Third Hospital, Beijing 100191, China; lelia_lj@163.com; 3Department of Health Research Methods, Evidence and Impact, Faculty of Health Sciences, McMaster University, Hamilton, ON L8S 4L8, Canada; 4Institute of Basic Research in Clinical Medicine, China Academy of Chinese Medical Sciences, Beijing 100700, China; zbzhangxiaoyu@aliyun.com

**Keywords:** Chinese herbal supplements, herbal tea, tea, drinking frequency, ADL disability, older adults

## Abstract

The aim of this study was to explore whether drinking herbal tea and tea would positively benefit activities of daily living (ADL) in the elderly. We used data from the Chinese longitudinal healthy longevity survey (CLHLS) to explore the association. Drinking herbal tea and drinking tea were divided into three groups using latent class analysis (LCA): frequently, occasionally, and rarely. ADL disability was measured by the ADL score. Multivariate COX proportional hazards models with competing risks were used to explore the impact of drinking herbal tea and tea on ADL disability, statistically adjusted for a range of potential confounders. A total of 7441 participants (mean age 81.8 years) were included in this study. The proportions of frequently and occasionally drinking herbal tea were 12.0% and 25.7%, respectively. Additionally, 29.6% and 28.2% of participants reported drinking tea, respectively. Multivariate COX regression showed that compared with rarely drinking, frequently drinking herbal tea could effectively reduce the incidence of ADL disability (HR = 0.85, 95% CI = 0.77–0.93, *p* = 0.005), whereas tea drinking had a relatively weaker effect (HR = 0.92, 95% CI = 0.83–0.99, *p* = 0.040). Subgroup analysis found that frequently drinking herbal tea was more protective for males under 80 years old (HR = 0.74 and 0.79, respectively), while frequently drinking tea was somewhat protective for women (HR = 0.92). The results indicate that drinking herbal tea and tea may be associated with a lower incidence of ADL disability. However, the risks associated with using Chinese herb plants still deserve attention.

## 1. Introduction

Disability in activities of daily living (ADL) is an important sign of aging in the elderly population [1,2]. Once ADL disorder occurs in the elderly, their health status will deteriorate rapidly. Globally, the prevalence of ADL disability among centenarians in the elderly population ranges from 61% to 81% [3,4,5]. However, in China, the proportion of individuals aged 65 and above is currently at 46% [1], which is expected to increase further due to population aging. If it is possible to delay the onset of ADL disability in the elderly population through lifestyle changes, it would be a highly cost-effective intervention. For example, it is worth exploring whether changes in dietary patterns can potentially reduce the incidence of ADL disabilities. Additionally, investigating the potential benefits of additional nutritional supplements to improve health status and further reduce the occurrence of ADL disabilities is also important. More and more studies are exploring the health effects of additional nutritional supplements [6,7,8], including extra vitamins and minerals, herbal and plant extracts, protein supplements, dietary fiber, sports nutrition supplements, etc.

The theory of Chinese medicine holds that some herbs have the same effect as medicine and food. That is to say, healthy people can take such herbs, similar to eating other foods, and obtain good health effects from them. For example, Renshen (Panax ginseng), Gouqizi (Lycium barbarum), and Danggui (Angelica sinensis) are such Chinese herbal medicines. There are many studies on them to explore their health effects and mechanisms [9,10,11,12,13]. However, few of these studies focused on populations, especially those conducted in long-term follow-up cohorts. A cohort study by Yi et al. demonstrated that ginseng intake was associated with decreased all-cause mortality in older males, but no life-prolonging effect was observed in women [14]. Chiu et al. used data from Taiwanese nationals aged 65 years and older and found that tea intake was associated with lower levels of disabilities in both men and women [15]. Yao et al. reported that a lack of tea consumption was one of the predictors of functional disabilities in a cross-sectional study in China [3]. Apart from the studies mentioned above, few studies have explored the health effects of drinking herbal tea and tea on ADL disability.

In this study, based on the Chinese Longitudinal Healthy Longevity Survey (CLHLS) [16], we carried out a study on the health effects of drinking herbal tea and drinking tea on ADL disability and compared the magnitude of their effects. CLHLS is a large follow-up cohort focused on the health of the elderly that was established in 1998 and has been followed up to now. In our study, we classified drinking herbal tea and drinking tea by latent classification analysis (LCA). The LCA would help us make a more scientific grouping of drinking herbal tea and drinking tea. Furthermore, we used multivariable COX proportional hazards models with competing risks to explore whether drinking herbal tea and drinking tea would decrease the incidence of ADL disability in the elderly.

## 2. Materials and Methods

### 2.1. Study Population

The data used in this study were derived from four waves (2008, 2011, 2014, and 2018) of CLHLS. The CLHLS cohort was established in 1998 by the Center for Healthy Aging and Development Studies of the National School of Development at Peking University. It focused on surveying individuals aged 65 and above and tracking and investigating the factors influencing their health. Survey details have been described in previous studies [16,17,18,19]. The datasets presented in this study are openly available in CLHLS at https://opendata.pku.edu.cn/dataverse/CHADS (accessed on 10 June 2023). The CLHLS cohort covered half of the counties and cities in 23 out of 31 provinces in China (these 23 provinces accounted for approximately 85% of the Chinese population) and had good sample representativeness for the elderly population in China. The CLHLS was approved by the Ethical Review Committee of Peking University (IRB00001052–13074). All participants signed an informed consent form at the time of their participation. The research was performed in accordance with the Declaration of Helsinki.

Since the survey on drinking herbal tea began in 2008, this study included and analyzed the data from 2008 to 2018 of the CLHLS. For the 2008 survey, 16,954 respondents participated and were interviewed. Participants aged above 65 years old who did the survey of drinking herbal tea and drinking tea at baseline and were followed up at least one wave later were eligible to include in this study. We excluded 2723 participants without an assessment on drinking herbal tea and tea at baseline, 2966 participants with ADL disability at baseline, 788 missing data on ADL score, and 3036 participants lost to follow-up, leaving a final sample of 7441 participants (Figure 1).

### 2.2. Assessment of Drinking Herbal Tea and Tea

The drinking of herbal tea and tea was measured by the questions: “How often do you drink the Chinese herbal supplement infusions (such as Renshen, Huangqi, Gouqizi, Danggui, etc.)?” and “How often do you drink tea?” The Chinese herbal supplements involved Renshen (Panax ginseng), Huangqi (Astragalus membranaceus), Gouqizi (Lycium barbarum), Danggui (Angelica sinensis), et al. The answer options for both questions were 1 out of 5, which were “1 almost every day”, “2 not every day, but at least once per week”, “3 not every week, but at least once per month”, “4 not every month, but occasionally”, and “5 rarely or never”. 

In each follow-up survey, participants were asked about their current consumption and whether they consumed medicinal plants and tea when they were 60 years old. In order to robustly assess the subjects’ drinking habits, we employed latent class analysis [20,21] (LCA) to categorize the responses to the above questions into three categories: rarely drinking, occasionally drinking, and frequently drinking (Appendix A).

### 2.3. Assessment of ADL Disability

The participants were assessed for their ADL score in each follow-up survey. ADL is a widely used assessment tool in healthcare and geriatric care settings [22,23]. It was designed to evaluate an individual’s ability to independently perform essential everyday tasks necessary for self-care and functioning. ADL assessment focuses on six items: dressing, bathing, indoor transferring, toileting, continence, and feeding. The participants were asked whether they needed assistance with each of the six items mentioned above. If they indicated needing help with any items, they were defined as having an ADL disability.

### 2.4. Covariates

The covariates included in this study were sociodemographic characteristics, a healthy lifestyle, and comorbidities (self-reported). Sociodemographic characteristics encompassed age (divided into three groups: <80, 80–90, and ≥90 years old), gender, ethnicity, educational level (divided into three groups: illiteracy, primary school or below, and junior high school or above), area of residence (divided into three groups: city, town, and rural), and co-residence status (divided into two groups: with household members and alone or in an institution).

Healthy lifestyle factors included smoking (divided into two groups: never and current or former), drinking (divided into two groups: never and current or former), consumption of fresh fruits (divided into three groups: almost every day, occasionally, and rarely or never), consumption of fresh vegetables (divided into two groups: almost every day and occasionally or rarely), exercise (divided into three groups: current, former, and never), and self-rated health status (divided into two groups: good or very good and general or below).

Comorbidities included in this study were the following five chronic conditions: hypertension, diabetes, heart disease, stroke, and chronic obstructive pulmonary disease (COPD). The number of comorbidities was determined by a count of self-reported health conditions.

### 2.5. Statistical Analysis

The LCA was performed to identify the classes of drinking herbal tea and tea by the “poLCA” package in R software, and an empirical approach was used to determine the exact number of latent classes [24]. The process started with two classes and progressed up to six classes, evaluating the classification quality using the Bayesian information criterion (BIC), Akaike information criterion (AIC), likelihood ratio, and Chi-square goodness of fit. Additionally, the selected classes had to have enough observations to provide a representative sample of a population.

The data were presented as the mean ± SD (continuous variables) or frequency distribution and percentage (categorical variables). Inter-group comparisons were conducted using the Chi-square test, Fisher’s exact test, or trend Chi-square test. The hazard ratios (HRs) and 95% confidence intervals (CIs) were calculated using the Cox proportional hazard model. The Schoenfeld residual test was used to assess the proportional hazard assumption. Considering that mortality was an important competing risk factor for ADL disability, a competing risk model was used in the COX regression [25,26]. The coefficients were calculated by the “survival” and “cmprsk” packages in R software and adjusted for competing risks of death.

Four different COX regression models were used to assess the impact of herbal tea and tea on ADL. The base model (Model 1) was the univariate analysis model; Model 2 controlled for the sociodemographic characteristics (age, gender, ethnicity, education level, area of residence, and co-residence status); Model 3 additionally controlled for the healthy lifestyle factors; and Model 4 added comorbidities based on Model 3. The subgroup analyses were performed in different categories of age and gender. 

All analyses were performed using SPSS version 26.0 for Windows software (IBM SPSS Inc., Chicago, IL, USA) or codes developed using R version 4.2.1 (R Foundation for Statistical Computing). A two-tailed *p*-value < 0.05 was considered statistically significant.

## 3. Results

A total of 7441 participants over 65 years old were included in this study, with an average age of 81.8 ± 10.44 years old. Among them, 4332 were over 80 years old (56.8%), and 3485 were male (46.8%). Overall, only 12% of all the participants in the study drank herbal tea frequently, 25.7% drank it occasionally, and most (62.3%) rarely drank it. Among different age groups, the proportion of participants < 80 years old drinking herbal tea was higher than that of participants over 90 years old, and the difference between groups was statistically significant (*p* < 0.001). The male was also more likely to drink herbal tea than the female. Similarly, individuals with higher education levels and those living in urban areas had a higher proportion of drinking herbal tea (Table 1).

In terms of lifestyle, individuals who smoke or drink alcohol are more likely to drink herbal tea. Among them, there were statistically significant differences in the consumption of medicinal plants (drinking herbal tea) between those who drink alcohol and those who do not (*p* < 0.001). Individuals who loved to drink tea, enjoyed consuming fresh fruits, engaged in regular physical exercise, and self-reported good health status were more likely to drink herbal tea compared to their control groups, and these differences were statistically significant (*p* < 0.05). However, among individuals with comorbidities, those with a higher number of comorbidities tended to have a greater preference for drinking herbal tea (*p* < 0.001), which may be attributed to the perceived health benefits associated with consuming medicinal plants (Table 1). The differences in baseline characteristics of the participants among different drinking tea groups are detailed in Appendix A.

The incidence of ADL disability was 40% (2974/7441) in all participants. The individuals in the group who rarely drank herbal tea or tea had a higher prevalence of ADL disability than other groups (*p* < 0.05). The incidence of ADL disabilities increases with advancing age. Females had a higher incidence of ADL disabilities compared to males. Urban residents had a higher incidence of ADL disabilities compared to rural residents. Additionally, individuals who frequently eat fresh fruits or vegetables had a lower incidence of ADL disabilities. Surprisingly, current or former smokers and drinkers had a lower incidence of ADL disability, which might be due to survivor bias or immortality bias. Among participants with different comorbidities, the higher the number of comorbidities, the higher the incidence of ADL disability (Figure 2).

The results of the unadjusted competing risk COX model showed that, compared with the rarely drinking group, both occasionally and frequently drinking herbal tea helped reduce the incidence of ADL disability, with HRs of 086 (95% CI: 0.77–0.96) and 0.84 (95% CI: 0.77–0.91), respectively. Drinking tea also had the same effect. However, after adjusting for other confounders, the protective effect of the occasionally drinking herbal tea or tea group did not exist, and only the frequently drinking herbal tea or tea group still had a positive protective effect. Similar results emerged in models 2, 3, and 4. In model 4, the protective effect of frequently drinking herbal tea on ADL disability was about 15% (95% CI: 7–23%) compared to the rarely group, and the protective effect of drinking tea on ADL disability was about 8% (95% CI: 1–17%) (Table 2).

Based on model 4, subgroup analysis showed that occasionally drinking and frequently drinking herbal tea had protective effects on ADL disability in males (HR = 0.85 and 0.79, respectively). In contrast, the protective effect was smaller in females (HR = 0.91). Meanwhile, the protective effect was greatest in those <80 years old (HR = 0.74, 95% CI = 0.63–0.87), while it gradually weakened in older age groups. Unlike drinking herbal tea, the protective effect of frequently drinking tea was stronger in women (HR = 0.92, 95% CI = 0.84–0.99) (Figure 3).

## 4. Discussion

With the progress of population aging, the difficulties in performing activities of daily living (ADL) of the elderly are becoming increasingly serious, which not only has a great impact on the quality of life of the elderly but also brings a huge burden to society [1,2,27]. It is increasingly important to mitigate ADL disabilities in the elderly population through nutritional supplements or lifestyle changes. This study aims to explore the effect of Chinese herbal supplements and tea drinking on ADL disability in the elderly population. The research data was based on the longitudinal cohort study of CLHLS in China. This study found that the proportion of elderly individuals who frequently drink herbal tea was around 12%, while occasional drinking accounted for approximately 25%. However, the proportions of drinking tea were higher than those of drinking herbal tea. A univariate competing risk model analysis indicated that both drinking herbal tea and tea had significant protective effects against ADL disability, and these protective effects increased with the frequency of drinking. Multivariate COX analysis showed that frequently drinking herbal tea had a protective effect of about 15% against ADL disability, while frequently drinking tea had a protective effect of only about 8%. Subgroup analysis demonstrated that frequently drinking herbal tea had a greater protective effect on males and those aged below 80. In contrast, frequent tea consumption had a more obvious protective effect on females.

Chinese traditional medicine theory believes that traditional Chinese herbal supplements positively affect health promotion. In some regions of China, there is a widespread habit of consuming Chinese herbal supplements (including drinking herbal tea), and some research has shown that these habits have a high level of acceptance among the local population [28,29]. However, these health-protective effects from traditional Chinese herbal supplements are often considered preliminary because of the potential effectiveness of placebos [30] and the lack of validation from long-term, large-scale cohort studies. Our study used a long-term, high-quality follow-up cohort of the elderly in China (CLHLS) to explore the health effects of taking traditional Chinese herbal supplements. This study comprehensively analyzed traditional Chinese medicine supplements such as Renshen (Panax ginseng), Huangqi (Astragalus membranaceus), Gouqizi (Lycium barbarum), and Danggui (Angelica sinensis). The study’s results found that the above herbal supplements had a positive health-promoting effect. In the past, many studies have explored the impact of these herbal supplements on patients. The RCT study by Hamidian et al. showed that regular ginseng supplementation might enhance physical, social, emotional, and functional well-being in breast cancer patients [31]. The review by Tauro et al. also pointed out that ginsenosides exhibit excellent anticancer activity, potentially serving as promising clinical candidates for cancer treatment [10]. Based on the Shanghai Women’s Health prospective cohort study [32], regular ginseng use was associated with an 8% lower risk of all-cause mortality, and a significant dose-response association was observed between the duration of ginseng use and total mortality. However, their other study based on the same cohort showed that although regular ginseng use was mostly not associated with the risk of any site-specific cancer, it may be associated with the risk of liver and thyroid cancers [33]. Many reviews have also summarized the positive effects of ginseng in the treatment of Alzheimer’s disease [9], stroke [34], sarcopenia [35], non-alcoholic fatty liver disease [36], and tumors [37], which might be related to the involvement of ginseng in immunomodulatory properties [38,39], anti-inflammatory [39], the regulation of signaling pathways [9], and the regulation of autophagy [11].

Astragalus membranaceus is a Chinese herbal medicine widely used in traditional Chinese medicine practices. Some studies have shown that Astragalus membranaceus has immune-regulating effects, boosting the function of the immune system and promoting the body’s resistance against infections and diseases [40,41]. Meanwhile, Astragalus membranaceus had some antioxidant and anti-inflammatory effects [41,42,43]. Lycium barbarum and Angelica sinensis are both widely used herbs in traditional Chinese medicine (TCM). According to TCM theory, they are believed to have excellent health benefits and are commonly used in dietary therapy by the Chinese population. Lycium barbarum polysaccharides (LBPs) are the main bioactive component in Lycium barbarum fruit and have considerable health-promoting effects. Zhou et al. showed that the fruit of LBPs can confer anti-diabetic effects in mice by changing the gut microbiota and gut barrier [13]. As a potential prebiotic, LBPs not only contributed to improving the body’s immune function, obesity, hyperlipidemia, and systemic inflammation caused by oxidative stress but also played an important role in regulating the gut microenvironment, improving host health, and exerting targeted effects on the intestines [44]. Sun et al. showed that LBP1C-2 extracted from Lycium barbarum was found to alleviate age-related bone loss by targeting BMPRIA/BMPRII/Noggin, which might serve as a new type of botanical medicine to fight age-related osteoporosis [45]. Several studies have demonstrated that Angelica sinensis could improve rheumatoid arthritis and colitis by reshaping the gut microbiota and also prevent splenic injury and functional impairment by inhibiting oxidative stress and cell apoptosis [12,46,47]. It also showed therapeutic potential in ischemic stroke treatment by regulating inflammation and autophagy pathways [48].

This national cohort study found that drinking tea was associated with an 8–10% lower risk of ADL disability, similar to previous studies. Chiu et al. used data from the National Health Interview Survey, which enrolled 10,898 Taiwanese nationals aged 65 years and older, to examine the factors related to physical disabilities [15]. They found that tea intake was associated with lower levels of all types of disabilities in both men and women. Yao et al. reported that lack of tea consumption was one of the predictors of functional disabilities in a cross-sectional study of a large sample of Chinese centenarians, including 180 men and 822 women, conducted from June 2014 to December 2016 [3]. As a beverage consumed for thousands of years, numerous claims about the benefits of tea consumption have been made and investigated. A review summarized that tea (especially green tea) could influence psychopathological symptoms (such as reduction in anxiety), cognition (such as benefits in memory and attention), and brain function (such as activation of working memory seen in functional MRI) [49]. The effects of tea could not be attributed to a single constituent of this beverage. In previous studies, the beneficial effects of tea on function and health were observed under the combined influence of both caffeine and l-theanine. In contrast, separate administration of either substance was found to have a lesser impact. The recent convention of introducing phytochemicals to support the immune system or combat diseases is a centuries-old tradition [50]. Tea has become one of the significant components of nutritional support strategies to maintain health and reduce the risk of various malignancies. Tea was reported to have significant antioxidative, anti-inflammatory, antimicrobial, anticarcinogenic, antihypertensive, neuroprotective, cholesterol-lowering, and thermogenic properties. Besides the benefits to physical function, many studies reported that tea consumption was also associated with a lower risk of diseases such as cancer, diabetes, arthritis, cardiovascular disease (CVD), stroke, genital warts, and obesity [50,51,52]. Although there are controversies regarding the benefits and risks of tea consumption, the health-promoting benefits of tea outweigh its few reported toxic effects [50,53].

Interestingly, in this study, we found that drinking herbal tea had a greater effect on reducing the incidence of ADL disability than drinking tea (15% vs. 8%). Notably, only a few previous studies have directly compared the impact of these two types of tea on ADL disability. In our opinion, this might be related to the medicinal properties of herbal tea. The health effects of tea might be more favorable than those of general health care products. Therefore, herbal tea appeared to have better health benefits relative to regular tea. However, it is worth noting that these teas may also present health hazards. Studies have shown that drinking green tea might cause liver damage [54], and nutritional supplements mixed with herbal ingredients might also cause kidney toxicity or other adverse drug reactions [55]. Therefore, before taking herbal supplements and tea products, the possible health hazards should be fully evaluated, especially in vulnerable groups such as pregnant women, the elderly, and children.

The present study included several strengths. This study is the first to explore the effect of drinking herbal tea on ADL disability based on a long-term follow-up cohort in the elderly population, with data from the CLHLS cohort [17,19,56]. The CLHLS cohort has been established and followed up since 1998. It is operated and managed by a dedicated team, and the data quality is reliable. In this study, LCA analysis was used to group the drinking habits of herbal tea and tea so as to ensure the reliability of exposure grouping as much as possible. In the data analysis, the COX regression of the competing risk model was used, and confounders such as demographic characteristics, lifestyle, and comorbidities were adjusted, striving to obtain more reliable research results.

However, there are several limitations that should be noted. First, this study mixed the drinking of herbal supplements together for analysis, such as Renshen (Panax ginseng), Huangqi (Astragalus membranaceus), Gouqizi (Lycium barbarum), Danggui (Angelica sinensis), and so on. The reason was that the original follow-up cohort did not investigate the consumption of medicinal plants separately. Therefore, the analysis of this data could not achieve separate analyses of medicinal plants one by one. Second, in the analysis of drinking tea, different types of tea were not analyzed separately, such as green tea, black tea, scented tea, etc. This was also due to the fact that this information was not separated in the survey of the original data. Third, there were also deficiencies in the adjustment of confounding factors. That is, only five comorbidities of the participants were included in the analysis, but the treatment of these comorbidities (such as hypertension and diabetes) and other comorbidities were not considered. At the same time, whether the treatment of hypertension and diabetes met the standard was not considered in the data analysis. The survey did not collect such data, so it was not analyzed. Fourth, this study might also be subject to immortal time bias [53] and the healthy worker effect, as only individuals alive had the opportunity to continue drinking herbal tea or tea. Similarly, the presence of current or former smokers and alcohol drinkers in the study showing lower rates of ADL disability could also be attributed to the immortal time bias. However, the study also revealed a phenomenon where individuals with a higher number of comorbidities tended to prefer drinking herbal tea, indicating that the baseline health status of the herbal tea group might be relatively poorer. Nevertheless, we controlled for these factors in the multivariable COX regression analysis. Last but not least, due to the inability to analyze different herbal teas separately in this study, the health effects obtained could not be clearly attributed to a specific herbal tea. In future research, it will be necessary to investigate the roles of different medicinal plants separately.

## 5. Conclusions

Our study found that frequently drinking herbal tea and drinking tea had a certain protective effect on ADL disability in the elderly population, and frequently drinking herbal tea had a greater protective effect than drinking tea. Subgroup analysis showed that the protective effect of drinking herbal tea on ADL disability was more obvious among men and the elderly under 80 years old. Additionally, the protective effect of drinking tea was more obvious among women than among men. These findings still need to be further verified in some randomized trials in the future. It is worth noting that drinking Chinese herbal tea should not be carried out blindly, especially among pregnant women, because it may induce some adverse reactions such as liver and kidney damage.

## Figures and Tables

**Figure 1 nutrients-15-02796-f001:**
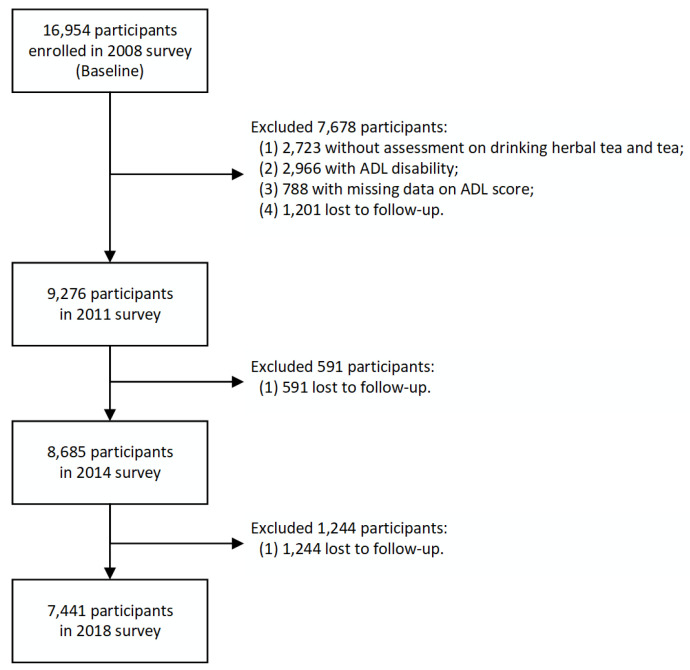
Flowchart of study participants.

**Figure 2 nutrients-15-02796-f002:**
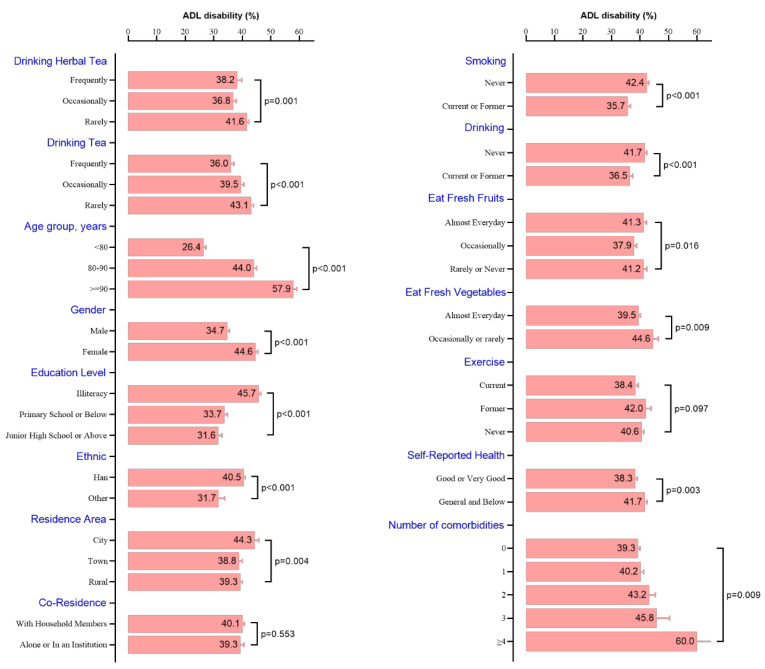
Associations between baseline characteristics and the incidence of ADL disability.

**Figure 3 nutrients-15-02796-f003:**
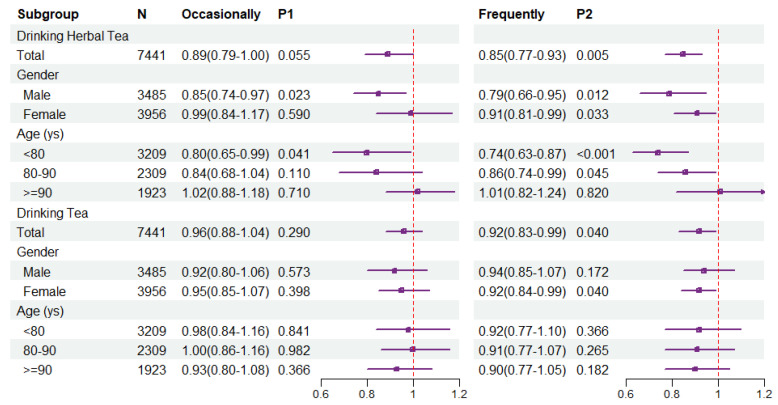
Subgroup analysis results of COX regression for drinking herbal tea and tea on ADL disability.

**Table 1 nutrients-15-02796-t001:** Baseline characteristics of the participants by different groups of drinking herbal tea (*n* (%)).

Characteristics	*n* (%)	Frequently	Occasionally	Rarely	*p*-Value
Total	7441 (100.0)	896 (12.0)	1911 (25.7)	4634 (62.3)	--
Sociodemographic characteristics					
Age group, years					<0.001
<80	3209 (43.1)	449 (14.0)	879 (27.4)	1881 (58.6)	
80–90	2309 (31.0)	274 (11.9)	593 (25.7)	1442 (62.5)	
≥90	1923 (25.8)	173 (9.0)	439 (22.8)	1311 (68.2)	
Gender					<0.001
Male	3485 (46.8)	473 (13.6)	902 (25.9)	2110 (60.5)	
Female	3956 (53.2)	423 (10.7)	1009 (25.5)	2524 (63.8)	
Education Level					<0.001
Illiteracy	4129 (55.5)	333 (8.1)	979 (23.7)	2817 (68.2)	
Primary School or Below	1887 (25.4)	239 (12.7)	556 (29.5)	1092 (57.9)	
Junior High School or Above	1425 (19.2)	324 (22.7)	376 (26.4)	725 (50.9)	
Ethnic					<0.001
Han	6964 (93.6)	867 (12.4)	1818 (26.1)	4279 (61.4)	
Other	477 (6.4)	29 (6.1)	93 (19.5)	355 (74.4)	
Residence Area					<0.001
City	1178 (15.8)	296 (25.1)	328 (27.8)	554 (47.0)	
Town	1599 (21.5)	218 (13.6)	497 (31.1)	884 (55.3)	
Rural	4664 (62.7)	382 (8.2)	1086 (23.3)	3196 (68.5)	
Co-Residence					0.275
With Household Member(S)	6058 (81.4)	745 (12.3)	1562 (25.8)	3751 (61.9)	
Alone or In an Institution	1383 (18.6)	151 (10.9)	349 (25.2)	883 (63.8)	
Healthy lifestyle and status					
Smoking					0.062
Never	4722 (63.5)	537 (11.4)	1216 (25.8)	2969 (62.9)	
Current or Former	2719 (36.5)	359 (13.2)	695 (25.6)	1665 (61.2)	
Drinking					<0.001
Never	4941 (66.4)	541 (10.9)	1269 (25.7)	3131 (63.4)	
Current or Former	2500 (33.6)	355 (14.2)	642 (25.7)	1503 (60.1)	
Drinking tea					<0.001
Frequently	2203 (29.6)	344 (15.6)	712 (32.3)	1147 (52.1)	
Occasionally	2095 (28.2)	292 (13.9)	624 (29.8)	1179 (56.3)	
Rarely	3143 (42.2)	260 (8.3)	575 (18.3)	2308 (73.4)	
Eat Fresh Fruits					<0.001
Almost Everyday	2895 (38.9)	484 (16.7)	828 (28.6)	1583 (54.7)	
Occasionally	2807 (37.7)	261 (9.3)	707 (25.2)	1839 (65.5)	
Rarely or Never	1739 (23.4)	151 (8.7)	376 (21.6)	1212 (69.7)	
Eat Fresh Vegetables					0.167
Almost Everyday	6732 (90.5)	826 (12.3)	1721 (25.6)	4185 (62.2)	
Occasionally or rarely	709 (9.5)	70 (9.9)	190 (26.8)	449 (63.3)	
Exercise					<0.001
Current	2588 (34.8)	481 (18.6)	696 (26.9)	1411 (54.5)	
Former	703 (9.4)	84 (11.9)	197 (28.0)	422 (60.0)	
Never	4150 (55.8)	331 (8.0)	1018 (24.5)	2801 (67.5)	
Self-Reported Health					0.004
Good or Very Good	3831 (51.5)	493 (12.9)	1019 (26.6)	2319 (60.5)	
General or Below	3610 (48.5)	403 (11.2)	892 (24.7)	2315 (64.1)	
No. of comorbidities					<0.001
0	4822 (64.8)	522 (10.8)	1254 (26.0)	3046 (63.2)	
1	1944 (26.1)	249 (12.8)	500 (25.7)	1195 (61.5)	
2	530 (7.1)	92 (17.4)	123 (23.2)	315 (59.4)	
3	120 (1.6)	22 (18.3)	25 (20.8)	73 (60.8)	
≥4	25 (0.3)	11 (44.0)	9 (36.0)	5 (20.0)	

**Table 2 nutrients-15-02796-t002:** Effects of drinking herbal tea and tea on ADL disability by competing risk COX regression.

	Rarely	Occasionally		Frequently	
		HR (95% CI)	*p*	HR (95% CI)	*p*
Drinking herbal tea					
Model 1	1 (Ref)	0.86 (0.77–0.96)	0.009	0.84 (0.77–0.91)	<0.001
Model 2	1 (Ref)	0.90 (0.80–1.01)	0.084	0.86 (0.80–0.94)	0.004
Model 3	1 (Ref)	0.89 (0.79–1.01)	0.064	0.86 (0.79–0.94)	0.006
Model 4	1 (Ref)	0.89 (0.79–1.00)	0.055	0.85 (0.77–0.93)	0.005
Drinking tea					
Model 1	1 (Ref)	0.89 (0.82–0.97)	0.007	0.84 (0.77–0.92)	<0.001
Model 2	1 (Ref)	0.95 (0.87–1.04)	0.231	0.91 (0.83–0.99)	0.036
Model 3	1 (Ref)	0.95 (0.87–1.04)	0.291	0.91 (0.83–0.99)	0.041
Model 4	1 (Ref)	0.96 (0.88–1.04)	0.290	0.92 (0.83–0.99)	0.040

Note: Model 1: unadjusted Competing risk Cox proportional hazards models. Model 2: Competing risk Cox proportional hazards models adjusted for age, gender, ethnicity, education level, area of residence, and co-residence status. Model 3 additionally controlled smoking, drinking, eating fresh fruits and vegetables, and exercising based on Model 2. Model 4 additionally controls for the number of comorbidities based on Model 3.

## Data Availability

The raw data supporting the conclusions of this article can be found here: https://opendata.pku.edu.cn/dataverse/CHADS (accessed on 10 June 2023).

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
