# Peer review of "Association of Drinking Herbal Tea with Activities of Daily Living among Elderly: A Latent Class Analysis"

_nutrients, 2023, doi:10.3390/nu15122796_

Round 1

Reviewer 1 Report

Dear Authors:

Regarding the manuscript with title “Association of drinking herbal tea with ADL disability among elderly: a latent class analysis, I have some major comments to address. Also several minor comments were posted in this review.

Major Comments:

Comment 1:

Line 63: “of healthy elderly”. How have you determined the health of the elderly people that participated on CLHLS? I do not understand this question as on Table 1 authors present data of elderly with self-reported health in the categorie of “General or below” and presente data of elderly with some comorbidities.

Comment 2:

Lines 89-95. The count from the respondents of the survey of 2008 until the final sample of this study is not correct. I ask authors to check the number of participants excluded and the reasons for that. Authors must insert a diagram regarding this point.

Comment 3:

Lines 97-98: “How often do you drink the Chinese herbal supplements infusions?” and “How often do you drink tea?” Do people know the difference between herbal tea and tea? How are authors sure of that?

Comment 4:

Lines 101-103: “1 almost every day”, “2 not every day, but at least once per week”, “3 not every week, but at least once per month”, “4 not every month, but occasionally”, and “5 rarely or never”. The 3 categories “rarely drinking”, “occasionally drinking”, and “frequently drinking” are related to which of the previous options (1 to 5)?

Comment 5:

On Table 1, why authors only present baseline data referring drinking herbal tea and do not presente data regarding drinking tea?

Comment 6:

Authors have to discuss why drinking herbal tea had a higher protective effect when compared to drinking tea. Authors have also to discuss regarding the results that found that frequently drinking herbal tea had a greater protective effect on the males and those aged below 80, whereas frequently drinking tea consumption had a more obvious protective effect on the females.

Minor Comments:

Comment 1:

On Title, I suggest authors to change “ADL” by “activities of daily living”

Comment 2:

Line 19: Regarding the following sentence: “relationship between drinking herbal tea and ADL disability”. Authors only assess this relationship or also the relationship between tea and ADL disability?

Comment 3:

Lines 28-29: I suggest authors to delete the following sentence: “However, the risks associated with the use of Chinese herb plants still deserve attention”.  Authors may discuss this theme on the chapter of Discussion.

Comment 4:

I suggest authors to change keywords to “Chinese herbal supplements; herbal tea; tea; drinking frequency; ADL disability; older adults”.

Comment 5:

Regarding the sentense “the prevalence of ADL disability among centenarians”: Why authors refer to people above 100 years old if the sample of this study was elderly people (65 years and above)? It has more sense to presente data from the prevalence of ADL disability among elderly people.

Comment 6:

Line 48: “Such as Renshen (Panax ginseng), Gouqizi (Lycium barbarum), Danggui (Angelica sinensis) etc., are such herbs”. Authors must correct the grammar of this sentence

Comment 7:

Lines 64-71: “In our study, we classified drinking herbal tea and drinking tea by latent classification analysis (LCA). LCA identifies unobserved subgroups based on response patterns. It assigns individuals to classes and estimates class probabilities[17], and increasingly used in social, psychological, and public health research[18–20]. The LCA would help us to make a more scientific grouping of drinking herbal tea and drinking tea. Furthermore, we used multivariable COX proportional hazards models with competing risks to explore whether drinking herbal tea and drinking tea would decline the incidence of ADL disability in the elderly.”. This information must be transferred from the chapter of Introduction to the chapter of “Statistical analysis”.

Comment 8:

The CLHLS cohort covered half of the counties and cities in 23 out of 31 provinces in China (approximately 85% of China's population). I do not understand the information relative to “85% of China’s population.

Comment 9:

Lines 125-126: Healthy lifestyle factors included smoking (divided into two groups: never, and now or ever). What means the expression “now or ever”?

Comment 10:

Lines 132-133: Participants only reported the 5 comorbidities written on these lines, correct?

Comment 11:

Line 226: “This study was to explore” by “This study aims to explore”.

Comment 12:

On final of Discussion, authors must refer that in future studies, it has to be investigated the role of different medicinal plants separately.

Minor editing of English language required. Some comments are regarding this question.

Author Response

Comments and Suggestions for Authors

Dear Authors:

Regarding the manuscript with title “Association of drinking herbal tea with ADL disability among elderly: a latent class analysis, I have some major comments to address. Also several minor comments were posted in this review.

Major Comments:

Comment 1: Line 63: “of healthy elderly”. How have you determined the health of the elderly people that participated on CLHLS? I do not understand this question as on Table 1 authors present data of elderly with self-reported health in the categorie of “General or below” and presente data of elderly with some comorbidities.

Reply:

Thank you very much for your inquiry. We made a mistake here, and we sincerely apologize for it.

Chinese Longitudinal Healthy Longevity and Happy Family Study (CLHLS-HF) collected longitudinal data coordinated by the Center for Healthy Aging and Development Studies of National School of Development at Peking University. It is a publicly available database and the website link is

https://opendata.pku.edu.cn/dataverse/CHADS;jsessionid=6c1211ed1ffb342615ece413d58c. The CLHLS-HF formerly known as the Chinese Longitudinal Healthy Longevity Survey (CLHLS), was renamed the CLHLS-HF in 2021. The current survey data for the year 2021 has not been publicly released yet, so we are still using the name CLHLS.

The CLHLS was a longitudinal healthy survey of elderly, and the elderly aged 80 years and over accounted for 67.4% of the total sample. So it is not a survey of healthy elderly individuals, but rather a survey focused on the health of the elderly. We also intended to convey this meaning in the article, but we apologize for the mistake due to language issues. We have now made the necessary corrections. Thank you again for bringing this to our attention.

Comment 2:

Lines 89-95. The count from the respondents of the survey of 2008 until the final sample of this study is not correct. I ask authors to check the number of participants excluded and the reasons for that. Authors must insert a diagram regarding this point.

Reply:

Thank you very much for your advice.

We apologize for mistakenly omitting the participants lost to follow-up in the original description. We have added a flowchart depicting the selection process of the participants. Please refer to Figure 1 for details. We deeply regret this error.

Comment 3:

Lines 97-98: “How often do you drink the Chinese herbal supplements infusions?” and “How often do you drink tea?” Do people know the difference between herbal tea and tea? How are authors sure of that?

Reply:

Yes, drinking Chinese herbal supplements infusions as part of daily life is a relatively common practice in China, and many Chinese individuals, especially the elderly, are familiar with it.

For herbal tea, the question in the original questionnaire was "How often do you drink the Chinese herbal supplements infusions (such as Renshen, Huangqi, Gouqizi, Danggui etc.)?” And for tea, the question was "How often do you drink tea?" Therefore, the participants could easily distinguish between herbal tea and regular tea.

We have also made the necessary revisions in the revised manuscript. Thank you.

Comment 4:

Lines 101-103: “1 almost every day”, “2 not every day, but at least once per week”, “3 not every week, but at least once per month”, “4 not every month, but occasionally”, and “5 rarely or never”. The 3 categories “rarely drinking”, “occasionally drinking”, and “frequently drinking” are related to which of the previous options (1 to 5)?

Reply:

We regret the previous unclear expression. Here, we used the latent class analysis (LCA) to classify the responses of the participants. After the evaluating the quality of classification, we finally divided the drinking habits of drinking herbal and tea into three categories, namely rarely drinking, occasionally drinking, and frequently drinking. The results of the LCA were shown in the table below, and we have also put this result in the revision (supplementary table 1).

Supplementary Table 1. The probabilities of drinking herbal tea and tea in Three-latent-class

Class 1

Class 2

Class 3

Drinking herbal tea

At aged 60 (2008 survey)

1 almost every day

0.063

0.001

0.000

2 not every day, but at least once per week

0.640

0.000

0.000

3 not every week, but at least once per month

0.174

0.000

0.005

4 not every month, but occasionally

0.123

0.989

0.054

5 rarely or never

0.000

0.010

0.941

2008 survey

1 almost every day

0.605

0.003

0.008

2 not every day, but at least once per week

0.131

0.000

0.024

3 not every week, but at least once per month

0.161

0.954

0.042

4 not every month, but occasionally

0.104

0.042

0.079

5 rarely or never

0.000

0.001

0.847

2011 survey

1 almost every day

0.654

0.001

0.071

2 not every day, but at least once per week

0.063

0.002

0.052

3 not every week, but at least once per month

0.080

0.011

0.068

4 not every month, but occasionally

0.151

0.928

0.166

5 rarely or never

0.053

0.058

0.643

2014 survey

1 almost every day

0.618

0.000

0.070

2 not every day, but at least once per week

0.103

0.003

0.060

3 not every week, but at least once per month

0.047

0.000

0.113

4 not every month, but occasionally

0.173

0.961

0.200

5 rarely or never

0.058

0.036

0.557

2018 survey

1 almost every day

0.062

0.005

0.072

2 not every day, but at least once per week

0.635

0.018

0.067

3 not every week, but at least once per month

0.081

0.934

0.059

4 not every month, but occasionally

0.147

0.032

0.140

5 rarely or never

0.075

0.010

0.663

Drinking Tea

At aged 60 (2008 survey)

1 almost every day

0.730

0.087

0.001

2 not every day, but at least once per week

0.095

0.155

0.013

3 not every week, but at least once per month

0.015

0.703

0.012

4 not every month, but occasionally

0.086

0.011

0.120

5 rarely or never

0.074

0.043

0.855

2008 survey

1 almost every day

0.802

0.144

0.000

2 not every day, but at least once per week

0.066

0.059

0.012

3 not every week, but at least once per month

0.019

0.778

0.014

4 not every month, but occasionally

0.059

0.010

0.107

5 rarely or never

0.055

0.009

0.867

2011 survey

1 almost every day

0.091

0.211

0.105

2 not every day, but at least once per week

0.757

0.086

0.052

3 not every week, but at least once per month

0.008

0.037

0.035

4 not every month, but occasionally

0.049

0.584

0.066

5 rarely or never

0.095

0.082

0.742

2014 survey

1 almost every day

0.732

0.069

0.073

2 not every day, but at least once per week

0.102

0.089

0.037

3 not every week, but at least once per month

0.021

0.744

0.028

4 not every month, but occasionally

0.040

0.056

0.054

5 rarely or never

0.106

0.043

0.808

2018 survey

1 almost every day

0.100

0.041

0.049

2 not every day, but at least once per week

0.589

0.040

0.034

3 not every week, but at least once per month

0.028

0.008

0.011

4 not every month, but occasionally

0.050

0.890

0.024

5 rarely or never

0.233

0.021

0.883

Note: Class 1 was defined as frequently drinking group, Class 2 was defined as occasionally drinking group, and Class 3 was defined as rarely drinking group.

Comment 5:

On Table 1, why authors only present baseline data referring drinking herbal tea and do not present data regarding drinking tea?

Reply:

Thank you for your suggestion. Initially, we thought that another baseline table based on drinking tea might be redundant. However, we have made the necessary modifications as your advice and added another table based on drinking tea in Supplementary Table 2.

Comment 6:

Authors have to discuss why drinking herbal tea had a higher protective effect when compared to drinking tea. Authors have also to discuss regarding the results that found that frequently drinking herbal tea had a greater protective effect on the males and those aged below 80, whereas frequently drinking tea consumption had a more obvious protective effect on the females.

Reply:

Thank you for your valuable suggestion. We have already added some discussion addressing this aspect in our discussions.

Minor Comments:

Comment 1:

On Title, I suggest authors to change “ADL” by “activities of daily living”

Reply:

Thank you for your suggestion. We have made the modification in the title.

Comment 2:

Line 19: Regarding the following sentence: “relationship between drinking herbal tea and ADL disability”. Authors only assess this relationship or also the relationship between tea and ADL disability?

Reply:

Thank you very much for pointing out the error. We have made the necessary modifications.

Comment 3:

Lines 28-29: I suggest authors to delete the following sentence: “However, the risks associated with the use of Chinese herb plants still deserve attention”.  Authors may discuss this theme on the chapter of Discussion.

Reply:

Thank you for your suggestion. But we don't really want to drop this sentence in the abstract. Because in our opinion, we still need to pay attention to this issue. After all, the possible health hazards of drinking Chinese herbal tea are not yet clear, and the potential liver and kidney toxicity of some traditional Chinese medicines has been widely pointed out.

Comment 4:

I suggest authors to change keywords to “Chinese herbal supplements; herbal tea; tea; drinking frequency; ADL disability; older adults”.

Reply:

Thank you very much for your suggestion. We have made the modification in the keywords.

Comment 5:

Regarding the sentence “the prevalence of ADL disability among centenarians”: Why authors refer to people above 100 years old if the sample of this study was elderly people (65 years and above)? It has more sense to presente data from the prevalence of ADL disability among elderly people.

Reply:

In the CLHLS cohort, the proportion of individuals aged 90 and above was relatively high. On its official website (https://opendata.pku.edu.cn/dataverse/CHADS), it also said that approximately 17% of the participants was centenarians, and approximately 23% were nonagenarians.

Comment 6:

Line 48: “Such as Renshen (Panax ginseng), Gouqizi (Lycium barbarum), Danggui (Angelica sinensis) etc., are such herbs”. Authors must correct the grammar of this sentence

Reply:

Thank you very much for pointing out the error. We have made the necessary modifications.

Comment 7:

Lines 64-71: “In our study, we classified drinking herbal tea and drinking tea by latent classification analysis (LCA). LCA identifies unobserved subgroups based on response patterns. It assigns individuals to classes and estimates class probabilities[17], and increasingly used in social, psychological, and public health research[18–20]. The LCA would help us to make a more scientific grouping of drinking herbal tea and drinking tea. Furthermore, we used multivariable COX proportional hazards models with competing risks to explore whether drinking herbal tea and drinking tea would decline the incidence of ADL disability in the elderly.”. This information must be transferred from the chapter of Introduction to the chapter of “Statistical analysis”.

Reply:

Thank you very much for your suggestion. We have made the necessary modifications.

Comment 8:

The CLHLS cohort covered half of the counties and cities in 23 out of 31 provinces in China (approximately 85% of China's population). I do not understand the information relative to “85% of China’s population.

Reply:

We apologize for the confusion caused by our unclear expression. The sentence intended to express that 85% of the Chinese population resided in the 23 provinces. We have made the necessary modifications.

Comment 9:

Lines 125-126: Healthy lifestyle factors included smoking (divided into two groups: never, and now or ever). What means the expression “now or ever”?

Reply:

We apologize for the confusion caused by our unclear expression. We have changed "now or ever" to "current or former".

Comment 10:

Lines 132-133: Participants only reported the 5 comorbidities written on these lines, correct?

Reply:

Thank you for your suggestion. Indeed, in the CLHLS cohort, not only these five comorbidities were collected, but also there have been changes in the collection of comorbidities across different years as the cohort study progressed.

However, in this study, we only included these five comorbidities. We apologize for the previous unclear expression, and we have made the necessary revisions in the revised manuscript. We have also addressed this limitation in the limitations section.

Comment 11:

Line 226: “This study was to explore” by “This study aims to explore”.

Reply:

Thank you very much for your suggestion. We have made the modification.

Comment 12:

On final of Discussion, authors must refer that in future studies, it has to be investigated the role of different medicinal plants separately.

Reply:

Thank you very much for your suggestion. We have made the necessary modifications in the Discussion.

Comments on the Quality of English Language

Minor editing of English language required. Some comments are regarding this question.

Reply:

Thank you very much for your suggestion.

Reviewer 2 Report

The sample you have studied is very high and significant. the interest in the study is also appropriate since the intake of these beverages is very common in this population.

The results are interesting, although not too relevant, since the improvement in the daily life of the subjects in the sample does not seem to be very conclusive. I think it's a good job

Author Response

Thank you so much.

Reviewer 3 Report

Interesting topic and associations. Well-written manuscript, good design, strong statistics, clear results and conclusion, recent references, and proper citations. It was good to mention the biases, and that it has to be made more investigations in this direction.

Author Response

Thank you so much.

Round 2

Reviewer 1 Report

Dear Authors.

All my comments were addressed, which improved the quality of this manuscript.